# "Only a God Can Save Us Now": Why a Religious Morality Is Best Suited to Overcome Religiously Inspired Violence and Spare Innocents from Harm

Alan Vincelette

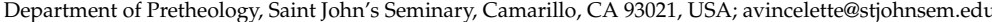

Department of Pretheology, Saint John's Seminary, Camarillo, CA 93021, USA; avincelette@stjohnsem.edu

**Abstract:** It is common to hear the refrain that religion is a major cause of violence today. And this claim is not without merit. Religious differences can fuel animosity and lead to societal conflict. On the other hand, scholars have increasingly recognized the role of religion in overcoming societal divides and helping people to heal and forgive. This paper will examine the latter capacity of religion to minimize the harms that occur during violent conflicts. It will be argued that secular ethical theories often fail to provide any principles or foundations that can help moderate passions, alleviate tensions, or provide frameworks for what is licit in war. In fact, the world views of terrorists and secular ethicists of war are often strikingly similar. Religious ethicists, on the contrary, have often encouraged practices (prayer for one's enemies, forgiveness) and provided principles (dignity of every human, non-combatant immunity, just war theory) that can help moderate the violent tendencies of war and bring about a more peaceful and equitable resolution. While religion is not entirely off the hook for promoting violent conflict, religion can provide ethical frameworks and principles that help minimize the harms of conflicts and promote world peace.

**Keywords:** war; religion; terrorism; revisionism; non-combatants; just war theory; peace

## 1. Introduction

The world is awash in religiously inspired violence, or so it seems. The first decades of the twenty-first century have witnessed the Second Intifada (2000–2005), the al-Qaeda attack on the World Trade Center (2001), the Houthi Insurgency in Yemen (2004–2014), the Barisan Insurgency in Thailand (2004–present), the Tehrik-i-Taliban Insurgency in Khyber Pakhtunkhwa, Pakistan (2004–2017), the Gaza Conflict (2008–present), the Boko Haram attacks in Western Africa (2009–2015), the Fulani attacks in Western Africa (2015–2023), and the Hamas attack on Israel (2023) among other conflicts (see Stern 2004; Juergensmeyer 2017).

I add "seemingly so" because there are complexities to the narrative that are often overlooked. Firstly, there is a danger of oversimplifying and stereotyping various revolutionary groups and their connection to religious motivations (Esposito 2003; Moritz and Mbacke 2022; Popal 2023). Secondly, motivations for these conflicts are complex and involve political and territorial disputes, persecution, and impoverishment in addition to religion disagreements (Esposito 1995; Cook and Allison 2007; Brym 2008; Canetti et al. 2010; Cavanaugh 2011; Coady 2013; Brubaker 2015; Wright and Khoo 2019; Coady 2021, pp. 176–206). In fact, quantitative studies have suggested that only seven to twenty-five percent of wars have religion as their primary motivating factor (Palmer-Fernandez 2003; Austin et al. 2004; Phillips and Axelrod 2004).

Still, religious beliefs are explicitly invoked by many terrorists as justification for their acts (Fox 2004; Wellman and Tokuno 2004), and so it is not surprising to see the blame for bloodshed placed on religion, in particular on Islam (Harris 2004; Avalos 2005; McCormick 2006; Ellens 2007; Steffen 2007; Persson and Savulescu 2013; Selengut 2017; Alcorta and Sosis 2022; Strathern 2023).

Now, if religion is the root cause of much political violence, then it would seem that we need to turn to secular ethical theories for the solution. Yet, not only are secular ethical theories lacking in their ability to resolve political conflicts, we find that many of them also advance views eerily similar to those of religious terrorists. Many secular ethicists, in particular so-called just war revisionists such as McMahan and Frowe, argue that violence is justifiable in extreme conditions and that non-combatants may be legitimate targets under certain circumstances.

On the other hand, while it is true that religious differences or beliefs can contribute to political conflict and war, it also turns out that religious ethics can provide many resources to minimize violence and resolve conflicts. For example, religions have invoked the importance of forgiveness and love of enemies, as well as human dignity and universal human rights, including the right of civilians to immunity from unjustified death. These resources of religion can be effective counters to religiously inspired violence if properly applied. The primary focus of this article will be on Western religion (in particular the Catholic and Anglican tradition best known to the author); however, other Christian traditions, as well as Judaism, Islam, and Eastern religions, will briefly be discussed.

This article will ultimately defend the importance of the principle of non-combatant immunity and its historical and metaphysical underpinnings in religion. After examining the justifications for attacks against non-combatants offered by leaders of terrorist organizations and secular ethicists, arguments in favor of non-combatant immunity will be produced. It turns out that many of the most skillful defenders of this principle are theists of one sort or another. And this makes sense as, if humans are made in the image of God and have dignity, then they must be respected and subjected to no unjustified harm. Far then from merely being the source of violence, religion has a lot to offer the world in its efforts to stem the tide of innocents harmed in religiously motivated conflicts. Heidegger was thus right in a way when he said "Only a god can save us now" (Heidegger et al. 1976), in spite of the fact that his own limited secular ethical theory was not strong and incisive enough to prevent him from affiliating with the National Socialists in Germany.

## 2. Religiously Inspired Violence

If we examine the statements made by leaders of terrorist groups, such as the Tamil Black Tigers of Sri Lanka, Hamas and the Al-Aqsa Martyrs' Brigade of Palestine, the Islamic Martyrs' Movement of Libya, Hezbollah of Lebanon, Al-Qaeda of Afghanistan, and the Islamic State of Iraq and Syria, we find that they often appeal to religious differences to countenance attacks on civilians as they reject a hard and fast distinction between combatants and non-combatants.

Osama bin Laden, the leader of Al-Qaeda, for example, in his Letter to the American People (2002), asserted that aggression against American civilians is justified since: (1) the American people freely choose their government, a choice that stems from their agreement with its policies, including support for the Israeli occupation of the land of the Palestinians; (2) the American people pay taxes that fund the military operations in the Middle East and through their elected candidates oversee the expenditure of these monies in the way they wish; (3) the American people employ their men and women in the armed forces in the Middle East; and (4) Allah allows the option of taking revenge and so whoever has destroyed Muslim villages and towns and killed Muslim civilians can have the same thing done to them (Bin Laden 2002). In other words, American citizens were appropriate targets of attack, since they were responsible for funding and empowering the persecution against Islamic Palestinians in the Middle East. Osama bin Laden also appealed to divine sanctions for revenge attacks: "Allah, the Almighty, legislated the permission and the option to take revenge. Thus, if we are attacked, then we have the right to attack back. Whoever has destroyed our villages and towns, then we have the right to destroy their villages and towns. Whoever has stolen our wealth, then we have the right to destroy their economy. And whoever has killed our civilians, then we have the right to kill theirs" (Bin Laden 2002).

Yousef Al-Qaradhawi of the Muslim Brotherhood likewise justified martyrdom operations against Israeli civilians by arguing that (1) every Israeli is a soldier in the army, either actually or because he or she can be summoned at any time into the army; (2) the Israelis are an invading and occupying populace on Palestinian lands and this does not diminish even with the passage of time; (3) Israelis are hostile to Muslims and so they annul the protection granted to the blood and property of believers; (4) just as it is permissible to kill innocent Muslims used as human shields to defend the Muslim community so too it is permissible to kill innocent non-Muslims in order to liberate the land of the Muslims from its occupiers and oppressors; (5) in modern warfare "all of society, with all its classes and ethnic groups, is mobilized to participate in the war, to aid its continuation, and to provide it with the material and human fuel required for it to assure the victory of the state fighting its enemies. . . . The entire domestic front, including professionals, laborers, and industrialists, stands behind the fighting army, even if it does not bear arms"; and (6) in times of extreme necessity one can engage in what is otherwise prohibited and so, in confronting the powerful weapons of the Israelis, the less well-armed Palestinians can use the weapon of suicide bombing as a countermeasure (Al-Awsat 2003).

Various leaders and spokespersons of Islamic Jihad have claimed that while they do not target children, elderly, schools, hospitals, or churches, other Israeli civilians are subject to attack as they are civilian occupiers not civilians in occupied territories; indeed, they are part of the occupying Israeli force as they are either armed settlers or potentially conscripted members of the Israeli armed services (Humans Rights Watch 2002, pp. 55–56). This line of reasoning was recently invoked by a spokesperson of Hamas, Osama Hamdan, who denied that Hamas had targeted civilians in their 2023 invasion of Israel on the basis that the Israeli settlers living in Palestine are not civilians but rather part of the Israeli occupying force (Aljazeera Explainer 2023).

### 3. Secular Ethical Theories

Seeing as many Islamic terrorists have appealed to religious motivations and political arguments in order to justify attacks against non-combatants (Kelsay 2007), it might be thought that the best way to counter their rhetoric is to turn to secular ethical theories. Surprisingly, however, when we do so, we all too often find viewpoints on the targeting of non-combatants quite similar to those proffered by terrorists. It is not that secular ethicists would necessarily support all of the actions of terrorist groups; indeed, they would likely find fault with many of their horrific acts of violence. However, secular ethicists are often in principle not opposed to attacks on civilians, sometimes on the same bases as those advanced by the terrorists themselves.

In particular, several ethicists working in public universities have developed what has come to be called a "Revisionist" just war theory, part of which maintains that non-combatant immunity in warfare should be more limited than traditionally thought and that sometimes acts of terrorism are permissible (Ignatieff 2005; Øverland 2005; Arneson 2006; McMahan 2009; Frowe 2014).

The most prominent revisionist is Jeff McMahan, who, since 2014, has been White's Professor of Moral Philosophy at Corpus Christi College, Oxford, following in the footsteps of W.D. Ross, H.A. Pritchard, R.M. Hare, and Bernard Williams. His book *Killing in War* (2009) argues that civilian immunity in wartime is not absolute and that "there are likely to be some occasions on which some civilians may be liable to intentional military attack, and on which it may be permissible to attack them" (McMahan 2009, p. 231).

McMahan draws a parallel between the permissibility of killing civilians in war and killing a conniving manipulator in self-defense. Suppose, hypothesizes McMahan, one's life is threatened by someone who has been told false stories, or been otherwise deceived, brainwashed, manipulated, or coerced, and that, furthermore, one can save one's life either by killing the directly threatening deceivee or the indirectly responsible deceiver. Since the deceiver in such a situation is the culpable cause of the threat, whereas the deceivee, even if the actual immediate threat is ultimately less culpable, McMahan argues that it would be

permissible, nay preferable, to save one's life through the killing of the deceiver instead of the deceivee. For, it is not the posing of an unjust threat that grounds liability to defensive harm (since one can pose a threat without being morally responsible for it) but rather *moral responsibility for an unjust threat* whether or not one currently poses such a threat (McMahan 2009, pp. 205–7, 227–28).

Applied to the ethics of war, this means that civilians can be liable to defensive harm if they are culpable causes of an unjust war, even if they are not currently physically endangering anyone. Indeed, there may well be situations where certain civilians are highly-culpable instigators, aiders, or abettors of an unjust war, while the soldiers fighting in it, being subject to misinformation or coercion, are less culpable. And, in such situations, it would be permissible, even preferable, to target and kill the civilians instead of the soldiers, presuming such a course of action would be just as effective in accomplishing one's military goals, and limit the overall harm done.

Now, this greatly extends the kind of civilian who can be subject to attack in war (McMahan 2009, pp. 210–12, 229–31). According to McMahan, civilians can be directly targeted in warfare when (1) there are present a number of unjust civilians who bear a substantial degree of moral responsibility for an unjust war; (2) attacking them would have a high probability of preventing the wrongful killing of an equal or greater number of innocents on the other side, whether civilians or soldiers; and (3) there are few, if any, genuinely innocent civilians nearby that would be harmed in such an attack.

McMahan gives a few historical examples of civilians who were permissible targets of violence. First are the executives of the United Fruit Company who, in 1954, lobbied the U.S. government to overthrow the democratically elected Guatemalan government as it had begun to nationalize some of the company's lands and to institute minimum-wage requirements. Second are the citizens of Pristina in Kosovo and Hiroshima in Japan, a sizeable majority of whom supported unjust acts of governmental aggression. And third are those very Israelis who were enthusiastic settlers on Palestinian lands in order to claim it for themselves and their offspring (McMahan 2009, pp. 214–16, 219–23, 228–30). Indeed, for McMahan, even those civilians who seek to arouse support for an unjust war through speeches or sermons; books, articles, or editorials; or lobbying political representatives are liable to being directly targeted, as are those civilians who acquiesce in their government's unjust actions and fail to protest or stop paying taxes, such as any Lebanese villagers who did not protest or prevent the launching of Hezbollah rockets into Israel (McMahan 2009, pp. 220–21; see also Dobos 2007; Bruenig 2010; Lamb 2013, pp. 42–45).

McMahan does stress that situations wherein civilians could permissibly be attacked would be "highly anomalous." For, most civilians bear only slight responsibility for a country going to war and have little capacity to affect the actions of their government after a war has commenced. Additionally, it is difficult to identify those civilians who bear a high degree of responsibility for an unjust war or to segregate them so they could be attacked without harming less responsible civilians. Finally, killing civilians is typically not an effective means of ending an unjust war. So, McMahan concludes "... that most unjust civilians are at most responsible to only a low degree for their country's unjust war, that attacks against civilians generally involve the opportunistic use of people as mere means, that they are virtually always of highly uncertain effectiveness because their relevant effects are not immediate but must come indirectly through the wills of others, that responsible civilians are virtually always intermingled with wholly innocent civilians—it is these factors together that explain why civilians are almost never liable to intentional military attack, and why even when some are liable it is still generally impermissible to attack them. These factors together constitute the real basis of the *moral* immunity of civilians, which has nothing to do with mere civilian status" (McMahan 2009, p. 231). McMahan additionally argues that it is typically in every country's best interest to adhere to the international laws prohibiting the intentional murder of civilians in war, for, if permission to attack civilians on occasion were recognized, it would very often be abused and a violation of a convention of war on one side would make it easier for the other side to lessen its commitment to

that convention. All together then, observes McMahan, "pragmatic considerations argue decisively for an absolute, exceptionless legal prohibition of intentional military attack against civilians" (McMahan 2009, p. 234).

More recent revisionist thinkers have pressed for even more extensive civilian liability in war and defended the permissibility of terrorism at times. Helen Frowe of the University of Stockholm, in her book *Defensive Killing* (Oxford: Oxford University Press 2014), aligns with McMahan in asserting that directly threatening unjust aggressors and indirectly threatening unjust agents are both liable to defensive harm if necessary. Hence, one can use potentially lethal force to defend oneself against someone unjustly attempting murder (a directly threatening unjust aggressor) or against someone willingly driving a car full of unjust mafia gunners (an indirectly threatening unjust accomplice). That is to say, indirectly threatening agents are liable to defensive harm to the extent that they are morally responsible for a threat (Frowe 2014, pp. 162–71).

Corresponding to this, it is licit in warfare for just combatants to attack indirectly threatening non-combatants when they are morally responsible for being unjust indirect threats. Akin to more traditional forms of just war theory, Frowe argues that non-combatants can become morally responsible indirect threats by designing, testing, or producing military weapons such as guns or bombs, military equipment such as jeeps or parachutes, or information technology utilized in warfare such as the software used to write or break codes. Frowe, however, goes beyond tradition in arguing that the British Women's Timber Corps, who harvested lumber during World War II, and the British Women's Land Army, who assisted in agricultural production after the German blockade, could be legitimate targets of attack as they supplied goods important to the war effort and freed-up male workers to join the battle front. For Frowe, even something as seemingly non-threatening as supplying clothing, food, or medicine to troops can make one complicit in an unjust war and subject to defensive harm. Finally, as with McMahan, non-combatants can become morally responsible indirect threats, and thus liable to defensive harm, by settling on disputed territories, voting for a candidate with a pro-war agenda, producing pro-war literature, or attending pro-war rallies (Frowe 2014, pp. 162–74, 185–86, 201–4, 210). Frowe, in fact, greatly extends the number of civilians who could be legitimately attacked in an unjust war. She argues that individuals failing to stop paying taxes, to join a march or civil protest, or to quit one's job in order to hinder an unjust war effort when they could do so with little consequence, are subject to defensive harm. Frowe, in fact, places a great deal of responsibility on civilians to investigate the justness of any war their country is involved in and to actively resist it if they determine it is unjust, unless doing so would be highly costly. She thus proposes that the typical citizen who contributes to an unjust war via standard forms of political involvement can only escape liability to defensive harm if there is "very good evidence that she lacks a reasonable opportunity to avoid posing that threat" (Frowe 2014, p. 187; see also Draper 2016, pp. 220–26).

Frowe, like McMahan, does distinguish between liability and, all things considered, permissibility, and professes that only rarely will liable non-combatants be permissible targets of attack. This is because there is great difficulty in identifying which non-combatants have responsibly contributed to the war effort (the identification problem), as well as in harming them without causing disproportionate collateral harm to non-culpable civilians (the isolation problem). For example, it would be very difficult to determine which civilians in a town worked at a military plant or voted for a war-mongering politician. Nor would it be easy to target the responsible civilians thus identified without harming the non-responsible ones, as responsible civilians tend not to live apart from non-responsible civilians, including innocent children. Finally, there will seldom be a military advantage for targeting such culpable non-combatants, since their contributions to the overall war effort tend to be small, they could easily be replaced, and the cost required to identify them, determine when they would be congregated apart from non-responsible parties, and launch a military attack on them would likely outweigh any benefit from their resulting deaths. Still, in the end, for Frowe, there is no absolute prohibition against directly attacking

civilians. Non-combatants are sometimes morally responsible for unjust lethal threats and so liable to defensive killing if it serves the purposes of a just war (Frowe 2014, pp. 195–98, 210–12; see also Palmer-Fernandez 2000; Steinhoff 2000; Slim 2008).

A few revisionists have even defended the legitimacy of terrorism in certain circumstances (Honderich 2003; Corlett 2003; Steinhoff 2007; Schwenkenbecher 2012). One of the most prominent is Virginia Held of the City University of New York, who wrote the book How Terrorism Is Wrong: Morality and Political Violence (2008). In it, Held argues that not just civilians who vote for a politician on the basis of a pro-war platform but even those who vote to elect and support politicians who wage an unjust war share responsibility with them for their policies and are legitimate targets of attack. For, if violence against the members of a state's army is justified, so too is violence against voting publics who put into power the governmental leaders who institute the unjust policies that harm other nations. Indeed, soldiers are often more innocent than said civilians, as they may be conscripted, misled into joining the armed services, or join out of necessity or when very young. Hence, Held queries "Is violence that kills young persons whose economic circumstances made military service seem to be almost their only option very much more plausibly justifiable than violence attacking well-off shoppers in a mall, shoppers whose economic comfort is enjoyed at the expense of the young persons who risk their lives in order to eat and thereby carry out the policies of the shoppers? It is hard to see here a deep moral distinction between combatant and noncombatant" (Held 2008, p. 78).

Held goes so far as to defend certain forms of terrorism against culpable civilians and proclaims "Instead of considering terrorism always and inevitably unjustifiable because it targets civilians, we should consider the aims of terrorists and of those who use violence to thwart those aims. We should compare the justice of the objectives of both sides, and we should compare the civilian casualties that both sides cause. The distinction between deliberately killing civilians and 'unintentionally' but entirely predictably doing so is of very limited moral significance" (Held 2008, p. 57). For all that, Held agrees there should be strong prima facie opposition to direct attacks against civilians and, consequently, justifying such attacks demands a great burden of proof, and such attacks may only be used as a last resort in order to achieve a more just distribution of rights violations.

Of course, not all secular ethicists would broaden non-combatant immunity so drastically, and many of them would only allow civilians to be targeted in cases of supreme necessity where one's fellow civilians face wholesale enslavement or extermination (Walzer 1977, pp. 251–68; Primoratz 2000; Steinhoff 2007, pp. 67–71, 93–97, 130–36; Fabre 2012, p. 253; Shue 2016, pp. 256–63). For example, Cécile Fabre, of All Souls College, Oxford, in her *Cosmopolitan War* (2012), asserts:

> The act of killing an innocent person—which infringes his right not to be killed and thereby extinguishes all his other rights—cannot be justified unless as a way to avert the greater evil of far greater numbers of individuals suffering a similar loss, or a violation, of all rights. . . . We must distinguish between a war in which W, as a political community, is under threat of destruction qua such community in the sense that P would take full control of its institutions if successful, and a war in which W is under threat of destruction qua community via a genocide (be it carried out by acts of killing or mass starvation) or the mass enslavement of its individual members. In the latter case, but not in the former, there is some justification, on the part of W's leaders, for ordering the deliberate targeting of (considerably fewer) innocent non-combatants as the only way to stave off the threat. (Fabre 2012, p. 253)

Still, on the whole, much of the contemporary secular just war theory allows many situations in which civilians can be legitimately targeted in war; so much so, that sometimes statements of terrorists and statements of revisionists begin to sound quite alike. Even McMahan was slightly embarrassed by this fact (McMahan 2009, pp. 232–34). That is, both modern terrorists and revisionists argue that civilians can be legitimately targeted if they have settled on lands that are not their own (of course who has a right to a a particular

parcel can be subject to much dispute), if they fail to do enough to stop an unjust war, or if they provide material, political, or moral support for those fighting an unjust war. For such reasons, it is hard to see how modern secular just war theory is well-positioned to curb religiously inspired violence. For this, we, in fact, need to turn to religiously inspired ethical theories.

## 4. Religiously Inspired Responses to Terrorism and the Harming of Non-Combatants

We have seen that many secular just war theorists open the door to a significant amount of direct civilian targeting in war, for the immunity of non-combatants is not grounded upon any deep moral principle about the material or moral innocence of non-combatants but instead upon contingent features of their situation. And so, revisionists seem to be hedging their bets that, in most circumstances, attacks on civilians will not be beneficial in pragmatic terms; yet, this seems highly questionable.

Take, for instance, their view that it would be quite difficult to identify those civilians who are culpable supporters of a just war. In light of the availability of social media today, it does not seem farfetched to use Facebook, LinkedIn, Twitter, Instagram, blogs, or other websites to find out a person's political views, what they are advocating, for whom they voted, or their place of employment. Presumably, if civilians were often directly targeted on the basis of what they wrote online, they might limit or eliminate such revelations, but, even so, some of this information might slip out or be available in past posts. A country could also employ spies, satellite imagery, hacking of computers and public video cameras, or paying people off for information in this endeavor. One could even imagine the use of artificial intelligence to identify neighborhoods in which all or nearly all the residents were enthusiastic supporters of a regime. So, identification of the political views of individuals might not prove too difficult.

Second, in terms of isolating culpable from innocent civilians, once culpable civilians were identified, a country could resort to assassination, poisoning, drone strikes, or other forms of execution to limit collateral damage. It could also employ satellite imagery and software to identify patterns of congregation of such individuals (see Toner 2004, p. 661; Downes 2008). So, situations in which civilians could permissibly be targeted would not seem to be the anomaly these thinkers argue it would be.

Doubtlessly, revisionists are correct in holding that attacking civilians is not often effective in ending a war, and indeed can harden the resolve of a country to continue to fight. Yet, if targeting civilians is considered licit in principle, then more countries or organizations could go that route in hopes of its effectiveness. Historically, arguments have been made on behalf of the strategic importance and effectiveness of obliteration bombing or dropping of atomic bombs on civilians by military leaders and politicians. Among the claims made were that targeting civilians can increase civilian unease with a war, lead to anti-war protests, changes in voting patterns, or pressure for politicians to end a war, and so aid in bringing a war to a halt.

See, in this regard, the support for obliteration bombing from the British General J.F.C. Fuller, who wrote:

> In traditional warfare, it was the rule that armies attacked armies and not non-combatants. If this tradition were strictly adhered to, then the demoralization of the enemy could only be effected by the destruction of the enemy's army and fleet. This process proved a most bloody one, and, during the war, adherence to it resulted in appalling slaughter. ... If, during the recent war [World War I], Germany could have been forced to disband her army and scrap her navy by a sudden and enormous loss of national morale, which entailed little blood-shed and small damage to her industries, would not the world today be more prosperous? ... And, supposing even if this sudden blow had cost the lives of a few thousand German women and children would such loss have rendered this novel type of warfare immortal? ... When, however, it is realized that to enforce policy, and not to kill, is the objective, and that the policy of a nation,

though maintained and enforced by her sailors and soldiers, is not fashioned by them, but by the civilian population, surely, then if a few civilians get killed in the struggle they have nothing to complain of (Fuller 1923, pp. 107–8; see also Sherman 1926; Spaight 1944; Wallace 1989; Garrett 2007)

Thus, there is a very slippery slope between advocating for increased targeting of civilians and more extreme terrorist measures. McMahan, Frowe, and Held do make it clear that they in no way support the indiscriminate killing of innocent civilians, but only morally culpable indirect threats. Yet, as this discrimination ultimately depends upon considerations of effectiveness more than a deep reluctance to harm innocents, its ability to call unwarranted attacks against civilians into question seems quite suspect, especially when terrorists advance similar sounding arguments.

Indeed, in times of war, more suspect motives, such as those for retaliation, may come to the fore and cause individuals to rationalize that bombing civilians is moral and an effective way to end a war. We already saw that along with utilitarian or ethical arguments, terrorists claimed (that they were only employing the violent measures against civilians that were already practiced on them).

This is where religious ethics can help prevent the death of innocents and moderate the worst harms of warfare, for religious ethics of war are based upon the idea that every human has dignity and is worthy of respect as they are created in the image of God (Gen 1:27; see Schlag 2013). Because every human has dignity, all humans must be respected no matter what their religion. If attacks are to occur, they must be severely restricted to unjust aggressors or militias or terrorists fighting an unjust war. Non-combatants must not be targeted (although there are some gray areas such as workers for companies providing direct material support for an army or industries producing weapons of war).

This principled opposition to the harming of innocent civilians is perhaps the greatest contribution of religion to social ethics (along with the notion that the highest form of love is the willingness to lay down one's life for another). Many Christian theologians have spoken out against indiscriminate targeting of civilians in war (Hartigan 1967). The Catholic ethicist John C. Ford wrote against the American and British policy of carpet bombing German cities in World War II (Ford 1944). So too did the Catholic convert G.E.M. Anscombe. She protested the granting of an honorary degree to American President Truman, as he ordered atomic bombs to be dropped on Japanese cities in order to shorten the war, and she also was horrified by the actions of the Allied forces against innocents such as flooding areas in the Netherlands full of innocent citizens in order to drive out the Nazis (Anscombe 1958). The Catholic encyclical *Gaudium et Spes* (1965) brings out this view: "Any act of war aimed indiscriminately at the destruction of entire cities or extensive areas along with their population is a crime against God and man himself which must be unequivocally and unhesitatingly be condemned" (n. 80). Such a condemnation of area bombing has also recently been enshrined in international law with Protocol I, Article 51 of the Geneva Convention (1977).

The Methodist ethicist Ramsey was equally opposed to any targeting of civilians in war. Ramsey voiced the following complaint:

> It is the concept of non-combatancy that has first been jettisoned from our minds; and this has happened because the concept of degrees of cooperation, the concept justifying the repulsion of objectively "guilty" forces as well as those "formally" or personally responsible for their direction, the concept of an indirect yet unavoidable and foreknown effect alongside the legitimately intended effects or military action, or the concept of double effects flowing from the same neutral or good action as cause, bringing along with the good result also a tragically necessary evil consequence in the limited, but not directly intended, yet foreseen destruction of civilian life (still not the same as wholesale murder, nor the same as a single murder)—all these notions have eroded from the minds of men. (Ramsey 1968, p. 156)

Such views, in fact, go back to Renaissance Scholastics such as the Dominican De Vitoria, who argued in his *De indis* (1532), nn. 36-38, that farmers, women, and children should be presumed innocent unless the contrary is shown and spared in warfare, since evil cannot be done to avoid future evils. Nor can one kill a child or adult who is likely to become an enemy soldier in the future, for it is intolerable to kill someone on the basis of a presumed future fault (Hartigan 1973; Eppstein 2008, pp. 432–55). In a similar manner, the Protestant Hugo Grotius, in his *De jure belli ac pacis* (1625), 3.11, claims that enemy civilians, including women and children, scholars, merchants, and artisans should not be directly attacked during a war (Little 1993; McKeogh 2012). These ideas are arguably based on such Scriptural texts that one should not do evil that good may come of it (Rom 3:8) and that one must love one's enemies (Mt 5:43–45). Or, in other words, that there are intrinsically evil acts that may not be carried out no matter the circumstances and one must be quick to show mercy and love. Such a principle has been powerfully defended in various Christian writings, in particular, Catholic and Anglican ones (Aquinas, *Summa theologiae*, I-II, q. 18, a. 4; *Catechism of the Catholic Church*, nn. 1750–1756; see McKeogh 2002; Kinsella 2011). Indeed prohibitions against the killing of innocent civilians in war are found throughout modern Jewish, Christian, and Muslim authors (Johnson 1971, 1975, 1981; Nardin 1998; Tansey 2004; Munir 2008).

Once we recognize the dignity of all humans and their right not to be unduly harmed, it becomes much harder to justify attacks against civilians. Focusing on the value of all humans also allows us to be more generous in not attributing complicity or responsibility to them regarding acts of war. In matters of war, clearly those who produce the weapons used by armed forces are cooperating in a close manner and so bear a high degree of culpability for an unjust war and can be targeted. However, those who farm the fields providing food for a nation, or vote for a politician, or provide moral support for a war are more remote cooperators and hence less accountable for a war effort and not subject to direct attack in warfare. Frowe is consequently wrong to assert that the British Women's Timber Corps and the British Women's Land Army who harvested lumber and farmed during World War II, and even Red Cross workers, bore responsibility and were licitly subject to direct attack for providing resources to soldiers (Frowe 2014, pp. 164–74, 202–10, 543–44), for their cooperation was remote, even if some of their wood or food ended up in military hands and their efforts freed-up more males to be soldiers. As Henry Davis observes:

> Non-combatants, i.e., those not engaged in actual aggression, nor under arms, nor in training, nor helping aggression, may not be directly attacked. The ordinary populace, going about their private business, children, youths under military age and not training are non-combatants. . . . Air raids on fortified towns, barracks, places of shelter for the forces, [and] munition factories, are permissible, but reasonable care must be taken, if possible, though usually this is impossible, to spare the lives and property of non-combatants. Indiscriminate air raids on non-combatants to sap the morale of a people are wrong. (Davis 1946, pp. 149–50; see also Biju 2015)

And the Anglican theologian O'Donovan also holds that the principle of distinction between combatants and non-combatants is of the utmost importance for the prosecution of a just war. He notes that it might be tempting to hamper the enemies' ability to pursue an unjust war by terrorizing his marketplaces or flattening his residential suburbs, but that "such a route to victory is one we should deny ourselves, since it denies the right of peaceful social existence, a right in which we and our enemy both share" (O'Donovan 2003, p. 40). And the Christian ethicist James Turner Johnson argues that the denial of a distinction between combatants and non-combatants is wrong, as people who do not directly participate in a war should not have the harm of war directed at them (Johnson 1999, p. 124).

Hence Cécile Fabre is wrong to claim that civilians who help to directly provide the military with food or healthcare are contributing to the war effort and are culpable and liable to targeting (even if in the end they should not be targeted for other reasons). She

writes "Although it is true that, strictly speaking, it is the guns as used by combatants which kill, not their specialized rations or wound dressing, it is equally true that combatants are not able to kill if hunger or untreated wounds make it impossible for them to lift their arms and train those guns on the enemy. Generally, meeting combatants' material need for food, shelter, appropriate clothing, and medical care goes a long way toward enabling them to kill in war, even if the resources in question do not in themselves constitute a threat" (Fabre 2009, pp. 43–44).

Fabre fails to discern the difference between proximate and remote material cooperation and how the former and not the latter generally make one responsible for the evil actions of those whom one assists. More astute here are the views of John Ryan and William Mattison. Ryan notes that:

> There are, however, degrees of cooperation; and even in the course of a long and bitterly contested war, it is impossible to admit that the vast bulk of the civil population has cooperated or even can cooperate so closely, either physically or morally, as to make them combatants. Women and children in the home, the aged, the sick, and the defective may lend the greatest moral support to both armies, but this does not make them combatants. The farmer, the baker, the tailor, the shop clerk, may all associate with and lend their services to the men and women in the armies, but, in the opinion of the present writer, their cooperation is too remote to make them actual belligerents. . . . If it is to be admitted that the entire civil population has become combatant in character and therefore subject to a completely devastating attack, it is because degrees of cooperation have lost all meaning, because sympathy for one's own has become the equivalent of physical opposition, and because a merely remote and potential danger can be reckoned as actual aggression. (Ryan 1940, pp. 110–11)

Similarly, for Mattison:

> Given the corporate nature of warfare, determinations of guilt or innocence which underlie designation as combatant or noncombatant must be based upon a person's particular activity, and the extent to which the activity directly contributes not simply to the war-makers (in terms of their health, stomachs, or moral support) but to their war-making activity. Munitions workers, pilots who shuttle soldiers to battle, and civilian contractors who erect an invading military's barracks may all be legitimate targets. But even with this recognition, surely a great majority of a nation's non-fighting persons remain noncombatants. (Mattison 2008, p. 170)

Revisionists such as McMahan and Frowe, as we have seen, also wrongly place a great deal of responsibility on the shoulders of civilians to investigate whether or not a war raged by their country is just or not and to actively resist it if it is unjust. For while it is easy to say in the abstract that such or such a war is just or unjust, in actual cases this is not easily determined. Wars typically involve territorial disputes that can go back generations, accusations of unjust treatment and acts of aggression, and other factors that are hard to understand or resolve. Hence, as the moral theologian Slater observes, "As a rule, international disputes are matters of great complexity, and it is very difficult to say on which side right and justice lie" (Slater 1925, p. 105; see also Jone 1955, pp. 136–37, 142–44). Even trained political philosophers have a hard time reaching a consensus on the justness of various policies and it seems clearly unjustifiable to say one can target a political philosopher on the opposite side of one's own in regard to the justness of a war. As Fellmeth puts it, "the justice of a resort to armed conflict is rarely black and white and never uncontested" (Fellmeth 2008, p. 464). Moreover, Coates notes that, as governments often need to withhold sensitive material that inform decisionmaking from the public, "The individual citizen is rarely in a position to make an informed and responsible judgement about the justice or injustice of the war . . . in the case of the individual citizen the moral presumption may be for war" (Coates 1997, p. 141), which, however, can be overcome

in the face of overwhelming evidence to the contrary. On occasion, governments even deliberately produce propaganda to deceive the public, again making their ability to judge the justness of a war much more difficult. Hence, civilians in general deserve a presumption of involuntary ignorance and should not be expected to come to a ready determination if a war is just or not.

Moreover, civilians are not wrong to place a certain level of trust in politicians and to recognize that, as civilians, they may not have all the facts on a matter. This is not to say that civilians should not seek to grapple with questions of justice regarding a war. Nor is this to say that in certain cases wars that are evidently unjust should not be opposed by civilians. It is just that, in matters such as politics, rational individuals can come to divergent conclusions. In the words of Koch and Preuss, "the individual citizen is rarely in a position to form a reliable judgment concerning the justice or injustice of a war . . . The causes of war are as a rule hidden to the ordinary citizen, nay oftentimes even to the better informed and more sensitive organs of public opinion. Ordinarily, the private citizen may and should presume that his country is right" (Koch and Preuss 1924, pp. 139–41). De Vitoria is again a precursor here and argues that citizens in a state may not have all the information necessary to make a declaration on the justness or not of a war (*De indis*, II, nn. 25, 30, 33). They should not, therefore, be held culpable for an unjust war.

It is on this basis that one may oppose the targeting of settlers on disputed territories. In the first place, such settlers, even if they bear arms (and even if they have violated an international law to not settle on occupied territories), are likely under the impression that they have a right to the land, and indeed may have been granted it by their government or provided with monetary inducements. Secondly, when such settlements become multigenerational, especially when over 50% of a settled population was born in a disputed territory or on colonized lands, it no longer makes sense to argue that the settlers or colonizers are not civilians but unjust invaders or part of a militia.

Of course, even within traditional Jewish, Christian, and Islamic circles, the line between civilian and non-civilian is not absolutely black and white (Tamer and Thörner 2021). For instance, those who worked in munitions factories were not considered civilians, or at least innocent ones, and so could be knowingly bombed if one was targeting the munitions factory at which they worked. Similarly, many Catholics held that political leaders could be targeted during a war (something international law seems less sure about). This is because there was a direct line between political leaders and the prosecution of an unjust war (Regan 1996, pp. 88–96). Various Catholics and Islamic moral theologians have also upheld the view that human shields need not stop one from attacking a military target, as the fault for killing civilians in such an attack lies with the enemy force (see, for instance, Jone 1955, p. 136, n. 211; Haque 2015; Munir 2011).

And there are some gray areas, such as whether one can target civilians who are war profiteers, who work in dual-use facilities such as oil refineries, those who shelter soldiers, or child soldiers and during siege warfare. Toner has argued that in addition to political leaders, loyal members of the ruling party who organize pro-war rallies or host recruiting functions, can also be legitimately targeted, especially during a supreme emergency, for such "noncombatant belligerents" are important contributors to a war effort and support soldiers qua soldiers not just qua human beings as farmers do (Toner 2004, pp. 654–55, 664). Toner asserts, albeit somewhat reluctantly, "The principle of graduated discrimination never licenses the targeting of the innocent. It does permit, however, in what can legitimately be called a supreme emergency (where the danger to a political community is imminent and grave), the direct targeting of any belligerents, even those who are very far from being actual combatants, not contributing in any material way to the enemy's war effort. This is because they are still contributing politically to the war effort by supplying the popular mandate for the regime and its military activities. They are helping to sustain the regime's 'will to fight'; this may be intangible, but it is far from inconsequential" (Toner 2004, pp. 658–59). That said, against Toner, the fact that a given populace, such as the Germans in World War II, was likely fed propaganda and false news, and that one can

generally trust one's leaders unless there is strong evidence to the contrary, would seem to minimize their accountability for an unjust war. It would be necessary to separate the well-informed yet still ardent Nazi supporters from those who were duped. And even then, I would not be comfortable with the targeting of civilians not holding key political positions. Targeting of a government official who merely speaks out in favor of a war does not seem warranted.

## 5. Religiously Inspired Ethics at the End of the War

Lastly, when a war is over, a religious ethics is well-positioned to help people move on and let former combatants find peace. At the conclusion of such conflicts, there are often a lot of residual feelings of hate and desire for revenge and temptations to incivility. There is a tendency to want to punish those against whom one has fought or those who cooperated with the enemy, i.e., collaborators (Waldron 2009). Christianity and other religions, however, have stressed that one should love even one's enemies (Mt 5:44), for all humans have dignity and make mistakes and it is necessary to forgive those who have done one wrong (Mt 6:14). Though there can and should be accountability and legitimate reprisals at the conclusions of a war, sometimes the punishments meted out seem exorbitant and based on feelings of revenge rather than justice. Religion here can assist healing after a war is over and indeed can be a great resource in the difficult task of "peace-building" (Aquino 2011; Powers 2020). As Coppola explains "Forgiveness is one key to freedom from the dehumanizing shackles of hate and revenge" (Coppola 2000, p. 41). Finally, as some have pointed out, even soldiers fighting in a just war may feel some guilt or experience trauma, and religion can aid in their healing by allowing opportunities for communal prayer and confession (Jackson-Meyer 2022). Religion then, in spite of what is sometimes alleged, can be a powerful force for peace, forgiveness, and healing in the world (Eppstein 1925; Appleby 2000; Kiess 2013; Sacks 2015; Burridge and Sacks 2018); such, in fact, is its proper end.

## 6. Further Considerations

Two issues, however, need to be addressed before we can conclude. First, this paper has principally focused on Western religion, and in particular the major Christian traditions that developed the just war theory, such as Catholicism and Anglicanism, though it has also brought in discussions of Judaism and Islam to some degree. Yet, what about other mainline or non-mainline Western and Eastern religious traditions? Would they also support the idea of non-combatant immunity? And can they also contribute to ending religious conflicts? That is to say, are there principles imbedded in religion that would tend to prohibit intentionally targeting civilians in war? My answer would be an unqualified yes to the first question, a qualified yes to the second, and a return to an unqualified yes to the third question.

Some of the Eastern religious traditions, such as Buddhism and Jainism, as well as a great proportion of non-mainline and mainline Western religions, tend to favor a pacifistic view and hold that nearly all wars should be avoided (Reichberg and Syse 2014; Jenkins 2023). Such a view is well-known in the Mennonite and Quaker "peace churches", whose theologians have written on the evils of war, including harm inflicted on innocent civilians, and have noted how religion can contribute to peace-building efforts (Yoder 1972; Friesen 1986; Lederach 1997; Wink 2003). Pacifistic viewpoints are also common among Jehovah's Witnesses, as well as mainline Methodists, Baptists, and Lutherans, and advocated by such theologians as Albert Schweitzer, Martin Luther King, Jr., A.J. Muste, Emil Fuchs, Dorothee Sölle, Stanley Hauerwas, Miroslav Volf, and Richard Hays (Schweitzer 1958; Hauerwas 1983; Sölle 1983; Volf 1996; Hughes 2008; Beaman and Pipkin 2013; Biggar 2013; Johnson 2023). And even though Hinduism can and has supported a just war framework (Hume 1916; Subedi 2003; King 2022a; Brekke 2023), it is perhaps today best known for the pacifistic teachings of Mohandas Gandhi (King 2022b).

Now, pacifism can be helpful in pointing out better approaches to the resolution of difficulties than resorting to war, and it can highlight the harms that occur even during just wars. Pacifists can thus model unconditional love and how to "embrace" even one's enemies (Volf 1996), and motivate bringing wars to an end. The challenge, however, is that pacificism can be inefficient in moderating violence, as it emphasizes the evil nature of all warfare and tends to plead for a particular war to end rather than contrasting legitimate and illegitimate tactics in war and encouraging the production of lesser over greater evils in the prosecution of war. Just war theorists, in this regard, are better able to latch on to what is actually going on in a war tactically and discriminate between different principles to be followed, as well as whom can or cannot be targeted. It is for this reason that Methodists such as Ramsey criticized statements made by the United States Methodist bishops (Ramsey 1990; see also Niebuhr 1940; Biggar 2013). In the end, though, all religions (or at least theistic ones), with the notion of the goodness of divine creation and the intrinsic value of human beings, have the resources to defend the dignity of all humans, their presumption of innocence, and their right to not be directly targeted in a war.

Second, are secular ethical theories doomed to failure in protecting civilians in war? Or, are there principles enshrined and resources found in modern liberal and secular ethics that can be drawn upon to also prevent unjust attacks against civilians in war?

While a large swath of modern secular ethicists take a revisionist position and loosen the prohibition against targeting non-combatants in warfare under certain conditions (albeit often unrealized), there are also secular ethicists who would oppose such a view. For example, Walzer distinguishes between those who provide food and those who provide weapons to an army and argues only the latter can directly be targeted as "It is not its belly but its arms that make it an army" (Walzer 1977, p. 146). Thomas Nagel similarly claims that "Contributions to [soldiers'] arms and logistics are contributions to this threat; contributions to their mere existence as men are not. It is therefore wrong to direct an attack against those who merely serve the combatants' needs as human beings, such as farmers and food suppliers" (Nagel 1972, p. 140; see also Murphy 1973; Sharp 1973; May 2007, pp. 167–89; Lazar 2015; May 2015, pp. 125–31; Miller 2016, pp. 185–211; Haque 2017, pp. 137–53, 263–65; Haque 2018; Coady 2021, pp. 81–109).

Indeed, there are various "secular" concepts of international law and human rights that have been enshrined in governmental military policies, as well as in international peacekeeping organizations such as Amnesty International, the Red Cross, and the United Nations, that can also undergird the protection of civilians. To take some examples, Article 51 of Protocol 1 for the Protection of Victims of International Armed Conflict of the Geneva Convention (1977) prohibits civilians from being objects of indiscriminate attacks, and this has been ratified in. the UK Government Strategy on the Protection of Civilians in Armed Conflict (2010) and the United States Department of Defense Civilian Harm Mitigation and Response Action Plan (2022) and Law of War Manual 5.5 (2023). The United Nations has developed its own Protection of Civilians mandate, pledging to do no harm to civilians and opposing indiscriminate attacks against them, and the United Nations provides personnel to monitor and minimize civilian harm and human rights abuses during conflicts (Willmot et al. 2016).

Ideally, religious and secular organizations should work together in peacekeeping efforts and in denouncing illegitimate attacks on non-combatants in warfare. Secular governments and international agencies have the material and monetary resources, personnel, and power to monitor, police, and help protect civilians during war, to provide information about human rights abuses, to provide funds for recovering from war, and to aid in remedying the unjust social structures that are often the causes of war. If the causes of war, as have been argued here, are often not in the main religions, then the cures of war must also involve secular governments and agents. All the same secular governments and agencies should continue to work with and provide roles for religious organizations and leaders in resolving conflicts and reconciliation. Religious institutions and leaders have intimate

knowledge of the sources of conflict, social networks in place to provide aid, and the power to influence their subjects (Funk and Woolner 2011; Haynes 2023).

Still, as we have seen, there are tensions in secular ethics and a tendency to loosen the protection provided to civilians in war, at least in many academics; it is not clear that secular governments and international agencies can, in the end, avoid doing so themselves (Slim 2008). Anscombe was not the first to point out the almost universal practice of obliteration bombing of cities during World War II upheld by American, British, Canadian, and German governments (Anscombe 1958). International law now follows her view but it is not clear that all governments will. Moreover, such secular natural rights theories originate from a religious natural law framework (especially from such Christian Scholastics as Vitoria, Suárez, Grotius, Pufendorf, Wolff, and Taparelli), and, although I cannot defend the idea here, can lose their force when completely divorced from this origin (Taparelli d'Azeglio 1855; Eppstein 1925, 2008; Tuck 1982; Tierney 1997; Janis and Evans 1999; Oakley 2005; Neff 2014; Vendemiati 2016; Domingo and Witte 2020; Slotte and Haskell 2021). Nor should the valuable resources of religious ethics, including the notions of forgiveness of enemies, the call to universal unconditional love, the weakness of human nature, and the seeing of all humans as valuable beings created in the image of God be overlooked as powerful forces for securing peace (Coward and Smith 2004; Haar 2005; Hertog 2010; Powers 2010; Huda and Marshall 2013; Omar and Duffey 2015; Omer et al. 2015; Garred and Abu-Nimer 2018).

## 7. Conclusions

Warfare continues to be an unfortunate part of life today. Such warfare often features conflict between people of different religions and, in many cases, religion provides some of the motivation for the conflict. Yet, situations are complex, and religion is often only one of several factors that lead to war, along with oppression or territorial disputes.

Moreover, religion has many resources that can be helpful in minimizing the harm inflicted on citizens in war and in securing peace at its end. In fact, one of the most important contributions of religion to restraining warring factions has been the principle of the immunity of non-combatants. Yet, this key principle, hammered out by various Christian just war theorists (and grounded in notions of human dignity as beings created in the image of God), has come under attack by revisionist theorists of the just war. Such revisionists (McMahan, Frowe, Held) have argued that discrimination between combatants and non-combatants is not in principle warranted, as non-combatants are often accountable for unjust wars, whether by cheering on an army, providing it sustenance, or voting in favor of the leaders likely to prosecute it. Hence, if civilians are to be protected in war, it must be on more pragmatic grounds.

Yet, there is no good reason to discard the principle of discrimination between combatants and non-combatants and indeed every reason to retain it. Pragmatic justifications of civilian immunity are more permissive of targeting civilians than often recognized, and only a principle of distinction that grants a fundamental difference between the lives of ordinary citizens going about their day to day activities and those actively engaged in a war can minimize harm to innocent civilians. The difference or distinction between actual combatants (or those directly assisting them) and civilians must be respected and non-combatants must not be directly targeted in war.

Moreover, if religion can be a partial cause of war, it can also be a major contributor to ensuring its just prosecution and fair and peaceful ending. When a war concludes, love of enemies, mercy, and forgiveness can help to ensure a peaceful resolution rather than a chance to punish enemies and allow vengeance to run rampant. So, there are many ways in which religion can be part of the solution rather than part of the problem when it comes to warfare. This, arguably, is one of its most important roles in society. In fact, the answer to limiting deaths of innocent civilians in wars and violent acts targeting them is not less religion but more an enshrining of a love for all at the center of one's mitigating and peace-building efforts.

**Funding:** This research received no external funding.

**Institutional Review Board Statement:** Not applicable.

**Informed Consent Statement:** Not applicable.

**Acknowledgments:** The author thanks comments by two anonymous reviewers, some of which have been incorporated here.

**Conflicts of Interest:** The author declares no conflict of interest.

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
