# Peer review of "“Only a God Can Save Us Now”: Why a Religious Morality Is Best Suited to Overcome Religiously Inspired Violence and Spare Innocents from Harm"

_religions, doi:10.3390/rel14121495_

Round 1
Reviewer 1 Report
Comments and Suggestions for Authors
I have very much enjoyed your article and found it clear, well argued and thought provoking. I note that you use the term 'religion' very generally and that you are predominantly writing from a Christian theological perspective and to some extent polarising secular ethics and religious ethics. You argue that secular ethical theories often fail to provide any principles or foundations that can help moderate passions, alleviate tensions, and provide frameworks for what is licit in war. However, there is a tendency to identify secular revisionist ethicists with the whole of secular ethics and I do recommend that you explore more widely the secular ethical theories underpinning international and national approaches to the protection of non-combatants. Please see my full review which is in the file attached. I am also pasting it below just in case!
‘The protection of civilians is a highly topical issue, which has been at the forefront of international discourse and taken a prominent role in many international deployments during recent years. Variously described as a moral responsibility, a legal obligation, a mandated peacekeeping task, and the culmination of humanitarian activity, it has become a high-profile concern of governments, international organizations, and civil society, and a central issue in international peace and security.’ Willmot, Haidi, and others (eds), Protection of Civilians (Oxford, 2016; online edn, Oxford Academic, 23 June 2016), https://doi.org/10.1093/acprof:oso/9780198729266.001.0001, accessed 16 Nov. 2023.
This article comes at a time when the protection of civilians is a matter of international debate in relation to the crisis in the Palestine and Israel, and when there has been an outpouring of prayers and statements on violence in Israel and Gaza by religious leaders and citizens all over the world. It is thought-provoking, raising many points of theoretical and practical importance. It is clearly written with a powerful central argument and message. The author argues that a religious ethics can help prevent death of innocents and moderate the worst harms of warfare. S/he defends the importance of the principle of non-combatant immunity and its historical and metaphysical underpinnings in religion. S/he examines the justifications for attacks against non-combatants offered by leaders of terrorist organizations and secular ethicists, and critiques them with arguments in favour of non-combatant immunity. S/he concludes that many of the most skilful defenders of this principle are theists of one sort or another. ‘And this makes sense as if humans are made in the image of God and have dignity then they must be respected and subject to no unjustified harm.‘ The author argues that not only are secular ethical theories lacking in their ability to resolve political conflicts, but that many advance views ‘eerily’ similar to those of religious terrorists.’ ‘Revisionist’ just war theorists maintain that non- combatant immunity in warfare should be more limited than traditionally thought and that sometimes acts of terrorism are permissible. The author sees theorists such as McMahan and Frowe, as opening the door to direct civilian targeting in war because the immunity of non-combatants is not grounded upon any deep moral principle about the material or moral innocence of non-combatants but instead upon contingent features of their situation. However, despite the differences noted by the author both revisionists and traditionalists on the whole agree that combatants should not deliberately harm non-combatants.
The author claims that modern secular just war theory lacks the resources to curb religiously inspired violence and to protect civilian non-combatants. On the other hand, s/he states that ‘For example, religions have invoked the importance of forgiveness and love of enemies, as well as human dignity and universal human rights including the right of civilians to immunity from unjustified death. These resources of religion can be effective counters to religiously inspired violence if properly applied’. In the last decades governments and international organisations have been increasingly willing to incorporate religion, to some degree, when addressing emerging challenges across a number of domestic and international policy areas. Religious elites often have a privileged role in conflict and reconciliation; they frequently possess a respected social status, extensive social networks, intimate understanding of the cultural-historical-regional context, and excellent communication skills. It therefore makes sense for the policy and diplomatic community to consider the benefits of engaging with religious organisations and leaders to fulfil the goals of security, justice, social cohesion and democracy. However, faith-based peace practitioners know that while religious peaceful concepts, narratives and rituals can mitigate situations of violence and hate speech, without attention to the structural or cultural violence which underpins war and conflict, just and lasting peace can seldom be achieved. Consensus needs to be built through negotiation with all parties to a conflict.
This article raises many questions not least about the author’s dualistic approach towards religious and secular ethics, but use of the term ‘religion’. Some scholars would argue that there is no transhistorical or transcultural concept of religion essentially separate from politics. They would argue that the attempt to say that there is a transhistorical and transcultural concept of religion that is separable from secular phenomena is itself part of a particular configuration of power, that of the modern, liberal nation-state as it developed in the West. Others would argue for the decolonisation of ‘religion’ and the fact that it trails predominantly Christian associations.
While the paper is fundamentally concerned with critiquing revisionist understandings of the just war argument, it would strengthen the argument if the author could give some attention to the ‘secular’ concepts of international humanitarian law and human rights which today underpin the protection of civilians. Individual states, the UN, regional organisations and humanitarian agencies including Non Governmental Organisations (NGOs) all play important roles in protecting civilians, whether through political and legal action, military activities or humanitarian action. The UK Government Strategy on the Protection of Civilians in Armed Conflict for example maintains that the protection of civilians is based on ethical, legal, humanitarian perspectives and international humanitarian law (IHL) which provides that civilians shall enjoy general protection from the effects of armed conflict, protects civilians from being the object of attack, and prohibits attacks that are indiscriminate. Similarly, the United Nations and the Protection of Civilians PoC has developed whole-of-mission approaches to promote protection as not only a military activity. Today, the Security Council regularly includes a whole host of activities in UN peacekeeping mandates for civilian and uniformed personnel to perform under the heading of PoC, including human rights activities, public information campaigns, preventative work alongside communities, quick impact projects, as well as robust uses of force to deter armed groups. The UN presents its mandate as based on an ethic of respect for human life and dignity. Civilians are not “collateral damage” and civilian harm is not an unavoidable consequence of conflict — civilian harm can and must be prevented. Armed actors are responsible and must be held accountable for preventing and addressing civilian harm.
The author might consider whether secular understandings of human and humanitarian rights and their enshrinement in international law can offer a transcultural language in which to oppose violence on non-combatants even as the resources that religious traditions offer can provide powerful incentives for peace. It should be noted that arguments based on non-violence whether instrumental (Gene Sharp and his successors) or on compassion or love (Gandhi, Dalai Lama) can be influential and efficacious. Gandhi’s militant nonviolence offers a possible alternative that avoids a complacent indifference toward injustice as well as the imitation of violence that leads to its escalation. Those who follow Sharp's pragmatic nonviolence approach believe in practicality rather than the moral aspect of the struggle.

Author Response
I thank the reviewer for the comments. I agree with the point that I should give more attention to secular ethics and organizations that can also be helpful in ending conflicts and helping to avoid deaths of innocent non-combatants. I have added a section [IN GREEN HIGHLIGHT] which looks at some of the principles coming from secular theories of natural rights and enshrined in important organizations such as the United Nations, which, as the reviewer notes, has been involved in many peace-keeping actions. I try to trace a bit more the strengths of such a natural rights approach and history. I then develop the idea of reviewer that cooperation of secular institutions and religious leaders can help in moderating violent conflicts.
Finally I try to incorporate a bit more some of the different religious views and traditions, such as the more pacifistic ones and Radical Orthodoxy and their relevance. [IN BLUE HIGHLIGHT]
Reviewer 2 Report
Comments and Suggestions for Authors
This is a fine and thoughtful paper with a compelling conclusion. What deserves more elaboration, however, is the difficult notion of religion. In other words, does each and every kind of religion-based ethics promote the same idea, i.e., that non-combattants ought not to be targeted intentionally during a war? Or are theism plus creationism presuppositions of such a position? In the paper, Christian, Jewish, and Islamic authors are quoted, all of whom probably subscribe to both theism and creationism. But how about other religious denominations?
Author Response
Thank you for your review and suggestions. I agree withy our point that more should be said on position of different religious traditions on the legitimacy of targeting non-combatants in a war. I have added a section [IN BLUE HIGHLIGHT] further detailing some additional religious traditions, their view and presuppositions.